# Potent and Broad but not Unselective Cleavage of Cytokines and Chemokines by Human Neutrophil Elastase and Proteinase 3

**DOI:** 10.3390/ijms21020651

**Published:** 2020-01-19

**Authors:** Zhirong Fu, Srinivas Akula, Michael Thorpe, Lars Hellman

**Affiliations:** Department of Cell and Molecular Biology, Uppsala University, Uppsala, The Biomedical Center, Box 596, SE-751 24 Uppsala, Sweden; fuzhirong.zju@gmail.com (Z.F.); srinivas.akula@icm.uu.se (S.A.); getmeinahalfpipe@gmail.com (M.T.)

**Keywords:** neutrophilic granulocyte, neutrophil elastase, proteinase 3, cytokine, chemokine

## Abstract

In two recent studies we have shown that three of the most abundant human hematopoietic serine proteases—mast cell chymase, mast cell tryptase and neutrophil cathepsin G—show a highly selective cleavage of cytokines and chemokines with a strong preference for a few alarmins, including IL-18, TSLP and IL-33. To determine if this is a general pattern for many of the hematopoietic serine proteases we have analyzed the human neutrophil elastase (hNE) and human proteinase 3 (hPR-3) for their cleavage of a panel of 69 different human cytokines and chemokines. Our results showed that these two latter enzymes, in sharp contrast to the two previous, had a very potent and relatively unrestrictive cleavage on this panel of targets. Almost all of these proteins were cleaved and many of them were fully degraded. In light of the proteases abundance and their colocalization, it is likely that together they have a very potent degrading activity on almost any protein in the area of neutrophil activation and granule release, including both foreign bacterial or viral proteins as well as various self-proteins in the area of inflammation/infection. However, a few very interesting exceptions to this pattern were found indicating a high resistance to degradation of some cytokines and chemokines, including TNF-α, IL-5, M-CSF, Rantes, IL-8 and MCP-1. All of these are either important for monocyte-macrophage, neutrophil or eosinophil proliferation, recruitment and activation, suggesting that cytokines/chemokines and proteases may have coevolved to not block the recruitment of monocytes–macrophages, neutrophils and possibly eosinophils during an inflammatory response involving neutrophil activation.

## 1. Introduction

Large amounts of immune mediators are found within cytoplasmic granules in cells of several of the major hematopoietic cell lineages. A large fraction of the proteins stored in these granules are serine proteases, which all belong to the large family of trypsin/chymotrypsin-related serine proteases [1,2,3,4,5]. Within this protease family many coagulation factors, complement factors and pancreatic digestive enzymes are also found. The members of this family that are expressed by hematopoietic cells have been named hematopoietic serine proteases. These proteases are primarily found in mast cells, neutrophils, natural killer (NK) cells and cytotoxic T cells, where they are stored in their active forms in cytoplasmic granules for rapid release. Very high amounts of these proteases are found in such cells, where the levels can reach 35% of the total cellular protein [6].

Neutrophils express four active serine proteases: N-elastase (hNE), proteinase 3 (hPR-3), cathepsin G (hCG) and neutrophil serine protease 4 (NSP-4), as well as one additional enzymatically inactive member of this gene family, azurocidine [1,4]. This latter protease homologue, has potent antibacterial activity but has, due to point mutations in two residues of the active site, lost its proteolytic activity [7].

Both hNE and hPR-3 are potent elastases cleaving after aliphatic amino acids such as Val, Ala and Ile, and with a relatively low preference for certain amino acids in their extended specificity. Interestingly hPR3 can also cleave after Thr [8,9]. This relatively low level of specificity results in a potential to cleave a large number of sites within most proteins. However, we have seen from a recent study of the human mast cell chymase (HC) that although potential ideal cleavage sites are found within a protein, it does not necessarily mean that the enzymes cleave at these sites. We have shown that this discrepancy primarily depends on accessibility of this target region. A site that is not presented on the surface of the protein may not be cleaved even if the site has a preferential amino acid sequence for the enzyme, whereas a relatively poor site when fully exposed may still be cleaved relatively efficiently [10]. In this recently published study of HC and hCG and their activity on a panel of 51 cytokines and chemokines, we showed that these two enzymes were remarkably restrictive in their cleavage properties. HC cleaved only four out of these 51 proteins efficiently, where the hCG targeted a few more but only then at an almost equimolar amount of enzyme to target substrate [10]. This showed that although sites may be present in a potential target they may not necessarily be cleaved, reflecting their highly selective behavior [10]. Among the most prominent targets for both of these enzymes were two alarmins, IL-18 and IL-33, indicating that these two enzymes may have major functions in regulating excessive inflammation [10]. This has been followed up by an analysis of the selectivity of another major mast cell enzyme, the tryptase. This enzyme was found to be even more selective. It did only cleave 2 cytokines and three chemokines out of a panel of 69 different cytokines and chemokines analyzed [11].

To look more deeply into this issue we have now focused on two of the most abundant neutrophil proteases hNE and hPR3, which are both elastases. These proteases have been shown to have a number of potential targets in vivo, and insufficient control of hNE activity has severe effects on lung function. Low plasma levels of one of the most important protease inhibitors controlling the activity of hNE, α1-anti trypsin has been found to result in severe lung emphysema. This emphysema is thought to primarily be caused by excessive cleavage of extracellular and cell surface proteins of the lung epithelium by hNE, which results in plasma leakage [12,13,14]. This phenomenon has indicated that these enzymes are very potent, which together with and a relatively low specificity results in major effects on the tissue or parasite target when released from the neutrophil in the area of infection or inflammation. In light of our recent finding of the quite restricted cleavage by hCG, we wanted to study other neutrophil enzymes in more detail to obtain a better insight in the biological function of these abundant enzymes. We therefore decided to study the efficiency of hNE and hPR-3 to degrade members of a panel of almost 70 recombinant human cytokines and chemokines. Our results showed that these two enzymes, in contrast to HC and hCG, have a very potent and relatively unrestrictive activity on this panel of proteins. Almost all the cytokines and chemokines were cleaved to some extent by both enzymes and the majority was almost completely degraded. Given the granule storage location and release, the four active serine proteases found in neutrophils likely also act together. We therefore analyzed a fraction of the cytokines and chemokines for the cleavage by a combination of three of these enzymes, hCG, hNE and hPR-3. An almost complete degradation of approximately more than 90% of these proteins was seen, indicating a potent and unrestrictive degradation of most target proteins by these enzymes in the area of infection/inflammation, which was in marked contrast from what we had observed for both the HC and the human tryptase which both appear to have more regulatory functions. Interestingly, a few cytokines and chemokines including TNF-α, IL-5, MCP-1, Rantes, IL-8 and M-CSF were remarkably resistant to cleavage, indicating that they may have been selected for stability against proteolytic degradation not to jeopardize the recruitment and activation of primarily monocytes/macrophages, neutrophils and eosinophils to the area of infection after neutrophils have entered the site of infection.

## 2. Results

### 2.1. The Proteases

The three neutrophil proteases used for this study were commercial preparations, where the original source of the protein was blood neutrophils (Figure 1). The same batches for all three of these enzymes have been analyzed for purity by SDS PAGE analysis and for activity on chromogenic substrate assay, phage display and a panel of recombinant substrates in a previously published study [15] (Figure 1). In this study we show that the enzymes are pure and that hPR3 and hNE both are elastases whereas hCG is a dual enzyme with both chymase and tryptase activity, the tryptase specificity with strong preference for Lys over Arg [9,15]. From the previous studies we have shown that hNE is much more active on a molar basis than both of the other enzymes. Therefore approximately 10 times more hPR3 enzyme was needed to obtain the same cleavage of chromogenic substrates compared to hNE [15]. Here we used 12 times more hPR3 during the cleavage reactions, which resulted in a corresponding band of approximately 28–29 kDa on the SDS-PAGE gels visualizing the reactions (Figure 2 and Figure 3).

### 2.2. Analysis of Sensitivity to Cleavage by hNE and hPR3

The purified hNE and hPR3 were used to study their cleavage activity on 69 commercially available active recombinant human cytokines and chemokines in vitro. The cleavage was performed in phosphate buffered saline (PBS) at physiological pH to try to mimic the in vivo situation under non-inflammatory conditions. To confirm the initial result, a new set of cytokines (a new batch) was used and the experiment was repeated under the same conditions as previously described. The results were consistent across all experiments. For the reactions, the enzyme to target ratio was approximately 1:50 when using hNE, whereas a ratio of 1:4 was used for hPR-3. It was apparent the stated amount of cytokine/chemokine from the supplier did not always match the observed quantity from the SDS-PAGE gels, therefore the ratio of hPR-3 to target varied from 5/6:1 to 1:1 (Figure 2).

Most of the 69 cytokines and chemokines were cleaved by both of these two enzymes and many were totally degraded by one of them or by both (Figure 2 and Table 1). Of the cytokines analyzed no cleavage or cleavage products were seen for only relatively few including several chemokines. Generally, the chemokines seemed to be more resistant to cleavage than the cytokines. This may be a reflection of the cytokines being larger in size and therefore probably more sensitive to cleavage. Due to the much lower activity of hPR-3 compared to hNE we used a higher concentration of the enzyme, which can be seen from the gels (the marked upper band is hPR-3, Figure 2). The cytokines and chemokines cleaved efficiently by both enzymes were TSLP, IL-13, SCF, IL-3 G-CSF, IL-2, IL-15, IL-20, IL-6, LIF, IL-7, IL-21, IL-16, IL-31, IL-19, IL-17A, IL-17F, IL-22, VEGF-A, PDGF-A, CTGF, FGF-1, FGF-9, IL-33, IL-18, IGF-1, IGF-2, sIL-6R, CNTF, HGF, FGF-19, GAL-7his, N-reg-1α, PF4V1 and TRAIL. A number were efficiently cleaved by hPR-3 but not hNE, these included IL-4, GM-CSF, IFN-γ, IL-12, IL-11, IP-10, MCP-3, SDF1α, SDF1β, MIP-1α, and TARC. TGF-β3 was the only one of cytokines and chemokines that was cleaved by hNE but not by hPR-3 (Figure 1). A few that were relatively resistant to cleavage by both enzymes included the cytokines IL-1-α, IL-1β, IL-5, M-CSF, IL-12, TNF-α, CD40-L, BAFF, PDGF-B, FGF-2, IL-1RA, BMP-14 and N-reg-1β, and the chemokines, EGF, IL-8, MCP-1, MCP-2 and RANTES.

### 2.3. Simultaneous Cleavage by Three Neutrophil Proteases, hNE, hPR3 and hCG

When neutrophils enter a tissue and release their granule content, the area around the neutrophil will experience a combination of at least four active serine proteases including, hNE, hPR-3, hCG and NSP-4. In order to study the effect on cytokine cleavage by a combination of these enzymes we tested a panel of the cytokines by simultaneous cleavage using three of these enzymes: hNE, hPR-3 and hCG. These three are the most abundant and the most active of the four neutrophil proteases and by assaying them together will thereby most likely give a good view of the total proteolytic activity of an activated neutrophil. The results showed a very potent degrading effect (Figure 3), where a complete or almost complete cleavage was seen for the following cytokines and chemokines: IL-15, IL-6, IL-17A, IL-2, IFN-γ, IL-18, IL-33, FGF-1, FGF-9, FGF-19, CTGF, IGF-1, IGF-2, MIP-1α, GRO-α, IP-10, CNTF, TRAIL, sIL-6R, TARC, sCD40-L, Neur-1α Neur-1β, pF4V1 and IL-19. Only TNF-α, FGF-2, BMP-14, EGF, MCP-1 RANTES and IL-1RA were relatively resistant to cleavage by the combination of three enzymes.

## 3. Discussion

In two recent studies of the HC, the human tryptase and hCG, we have shown that these three enzymes are very selective in their cleavage of a large panel of cytokines and chemokines. This result was in marked contrast to the previous dominating view of the hematopoietic serine proteases, which assumed a relatively unspecific nature and cleavage of almost any substrate if allowed to do so for extended periods of time. The HC only cleaved four cytokines and chemokines out of a panel of 51, which shows a high selectivity for this mast cell specific protease [10]. In this study hCG did cleave a few additional cytokines and chemokines [10]. However, similar to what we show here for hPR-3 and hNE, hCG has a much lower catalytic activity compared to the HC, therefore we used a higher concentration of hCG than for HC, which at least partly explains the higher number of cleaved targets [9,10,16].

The human tryptase was even more restrictive in its cleavage of 69 different cytokines and chemokines, only two cytokines, TSLP and IL21 and three chemokines MCP3, MIP-3b and eotaxin were efficiently cleaved by this enzyme [11]. We then became interested in the possibility that other hematopoietic serine proteases were more restrictive in their target selection than previously thought. Some of the hematopoietic serine proteases that have been considered the most active and unrestrictive in their activity are the neutrophil proteases. We therefore wanted to study their activity in more detail to obtain a better view of their activity against a larger panel of human potential substrates. Subsequently we selected a panel of 69 different human cytokines and chemokines. Here we have shown that these enzymes, in contrast to the HC and the human tryptase were relatively unrestrictive and cleaved the absolute majority of these potential targets. Furthermore, the combination of three almost completely degraded the majority of the tested cytokines and chemokines. This finding confirms the view that these enzymes are relatively unrestrictive and highly active in the area of neutrophil activation and granule release. They are most likely key players in degrading the surrounding extracellular matrix including connective tissue components enabling the neutrophil to reach the area of infection, and when they enter the site of infection they can then degrade bacterial virulence factors. At these sites neutrophils can potentially use extracellular traps to limit bacterial spread and provide time for efficient phagocytosis to finally remove the infectious agent. The three proteases analyzed together in this study, hNE, hPR-3 and hCG, are the most active, are the most abundant and have the broadest specificity of the four active neutrophil granule proteases. The forth active serine protease of neutrophil granules is NSP-4, which seems to be quite specific and likely similarly to HC, may have a yet unknown, regulatory function [9]. Due to the broad specificity of hNE, hPR-3 and hCG and the high number of aliphatic amino acids in almost all of these analyzed cytokines and chemokines the possibility to cleave would be almost 100%, if not folding did influence accessibility and thereby cleavage. The concentration of these enzymes in the area close to an activated neutrophil is probably also high indicating that cleavage could occur of cytokines and chemokines even in the presence of high amounts of connective tissue and blood proteins. These proteases are probably also the dominating proteolytic enzymes in the area of inflammation due to their abundance and high activity.

Interestingly, in addition to the general degrading effect we also found a few unexpected examples of cytokines that were quite resistant to degradation by these enzymes (Figure 2). What this tells us about the role of these cytokines is still not fully clear but may offer us important clues to the regulation of recruitment and maintenance of the inflammatory response. IL-1-α, M-CSF, IL-5 and TNF-α are four examples of cytokines that were quite resistant to cleavage by these enzymes, indicating that the tissue may need these to ensure efficient recruitment and activation of monocytes/macrophages and eosinophils in order to clear up after interaction between numerous neutrophils and bacteria, or as for the eosinophils to act on larger parasites (Figure 4).

This is a situation that may also apply for the various chemokines where a majority seems to be quite resistant to cleavage. The chemokines IL-8, MIP1, MIP3, and RANTES are examples of such relatively stable chemokines. They are key players in the recruitment of inflammatory cells to areas of infection and would therefore be beneficial to be relatively resistant in order not to be removed before a sufficient number of inflammatory cells have reached the area (Figure 4). It is also possible that they are more stable to resist cleavage by bacterial proteases. The high stability of the IL-1RA could also be important as it acts as an antagonist of IL-1α and β and regulates thereby the activity of these inflammatory cytokines and when IL-1α and partly also IL-1β are resistant to cleavage they need to be balanced not to induce excessive inflammatory responses. We also observed a high resistance to cleavage for the epidermal growth factor (EGF), PDGF-B and for BMP-2 and 14 indicating that factors of importance for the repair of epithelial tissues, blood vessels and bone following an infection/inflammation also are protected from cleavage by these neutrophil enzymes.

IL-32, a cytokine not tested in this analysis, has previously been shown to be bind to and to be cleaved by PR-3 [17]. The cleavage does in this case not reduce but instead increase the activity of IL-32 [17]. Interestingly this cytokine has been shown to induce the production of several cytokines and chemokines including IL-1β, IL-6 and TNF-α, indicating that it even in the presence of these neutrophil proteases this cytokine can contribute to the inflammatory response [17].

It shall here be noted that neutrophil proteases and neutrophil extracellular traps (NETs) have previously been shown to limit inflammation by the cleavage of cytokines and chemokines [18,19]. However, a more detailed analysis of a large panel of cytokines and chemokines has not been performed previously. Interestingly from the earlier studies, reactive oxygen species (ROS) seems to be important both for the release of proteases and the formation of the NETs [19,20]. Proteomics studies have also indicated that proteases constitute approximately 10% of the protein content of NETs [21]. There are also reports on the activation of several cytokines, including IL-1, IL-33 and IL-36, by the neutrophil proteases and thereby escalating inflammation [22,23,24]. These two processes may occur simultaneously and may depend on the number of neutrophils and their level of activation, and also the cytokine in question. Some more stable cytokines like IL-1 and possibly IL-36 may preferentially be activated, whereas the majority of cytokines will most likely be degraded. At higher enzyme concentrations, when all three proteases are present and in the presence of NETs the degradation is most likely the dominating activity. The precursors of IL-1, IL-33 and IL-36 are also present primarily in the cytoplasm of the cells, or the nucleus for IL-33, why the cells need to be damaged in order for the neutrophil proteases to be able to cleave. The IL-33 precursor does show some activity before cleavage, however, the activity increases by 30 times after cleavage [25]. Due to their cytoplasmic or nuclear localization activation by the neutrophil proteases may therefore at least theoretically primarily be a process of importance during tissue damage or necrosis. Partial trimming can also either inactivate or activate a cytokine or chemokine. One of the chemokines found to be relatively stable in this study, IL-8, has previously been shown to be activated by N-terminal cleavage by hPR-3 [26]. This N-terminal cleavage is also observed in Figure 2.

In order to perform all the different functions needed at the area of inflammation, the proteases stored and secreted by the neutrophils have relatively broad specificities both concerning their primary and extended specificities. We have recently performed detailed studies of the extended specificities of three out the four active serine proteases stored in granules of human neutrophils, hCG, hNE and hPR-3 [9,15]. Phage display analysis and validation by a large panel of recombinant protein substrates have shown that hCG is a dual chymase and tryptase with a dominating chymase activity against aromatic amino acids, and with a tryptic activity primarily against lysine that is only 2–3 times lower than the major chymotryptic activity. The hPR-3 and hNE are both broad specificity elastases with primary specificities against aliphatic amino acids including valine, isoleucine and alanine. Interestingly, as shown by cleavage of a library of peptides and subsequent analysis by cleavage products by mass spectrometry, hPR-3 also shows potent activity against substrates with threonine in the P1 position (the amino acid after which the enzyme cleaves) [8,9]. The only protease of the four active serine proteases that is still poorly characterized is also the most recently identified member, the NSP-4 [27,28]. This protease is found in relatively low concentration within neutrophil granules has been shown to have tryptic activity. However this activity is very low against all the chromogenic and recombinant substrates analyzed, indicating that the extended specificity is still largely unknown [9]. We have performed several phage display attempts to obtain more information of this interesting protease without much success, which gives additional indications that it is highly specific and thereby difficult to study even with a technique as sensitive and selective as phage display. The potential highly specific nature of NSP-4 may indicate a quite different function compared to the other three neutrophil serine proteases. It may have a regulatory function against a still unknown substrate. Furthermore NSP-4 seems to be one of the most highly conserved of all the neutrophil proteases during vertebrate evolution [5].

Given the data provided here and from several earlier studies we can now conclude that three of the neutrophil granule proteases seem to have relatively broad specificities aimed at cleaving both connective tissue components, bacterial virulence factors and other proteins, whereas the fourth, NSP-4 is most likely more restricted in its target specificity and instead may have regulatory functions. Interestingly was the observation of the relatively high resistance to degradation by a few inflammatory cytokines and the majority of chemokines indicating that they are selected for stability to ensure that the recruitment of inflammatory cells into an area of bacterial infection is not stopped or blocked by their degradation. These signals are most likely needed to be relatively intact even in the presence of the high proteolytic activity due to their essential requirement for recruitment, proliferation and activation of cells that are needed at the site of infection. A few cytokines and chemokines of importance for tissue repair was also resistant to cleavage, including EGF, PDGF-B, BMP-2 and 14, indicating that these are stabilized to resist degradation to enable repair of damaged tissues after an inflammation/infection.

## 4. Materials and Methods

### 4.1. Enzymes

Human proteinase 3, hNE and hCG were purified from peripheral blood neutrophils and purchased from Lee Biosolutions (St. Louis, MO, USA), Athens Research & Technology (Athens, GA, USA) and BioCentrum (Krakow, Poland), respectively. The same batches of enzymes have been analyzed for purity on SDS-PAGE gels and for activity by assay on a panel of chromogenic substrates in a previous publication [15] (Figure 1).

### 4.2. Recombinant Cytokines and Chemokines

Seventy two recombinant human (rh) cytokines and chemokines were purchased from Immuno Tools (Friesoythe, Germany), including rhActivin A active, rhBAFF/CD257, rhBMP-2, rhCNTF, rhCTGF, rhEGF, rhFGF-α/FGF-1, rhFGF-β/FGF-2, rhFGF-9, rhFGF-19, rhFlt3L/CD135, rhGAL-7His, rhG-CSF, rhGDF-5/BMP-14, rhGM-CSF, rhGROα/CXCL1, rhHGF, rhIFN-gamma, rhIGF-1, rhIGF-2, rhIL-1α/IL1F1, rhIL-1β/IL1F2, rhIL-1RA/IL1F3, rhIL-2, rhIL-3, rhIL-4, rhIL-5, rhIL-6, rhsIL-6R, rhIL-7, rhIL-8/CXCL8, rhIL-10, rhIL-11, rhIL-12, rhIL-13, rhIL-15, rhIL-16, rhIL-17A, rhIL-17F, rhIL-19, rhIL-20, rhIL-21, rhIL-22, rhIL-31, rhIP-10/CXCL 10, rhKGF-2/FGF-10, rhLIF, rhMCP1/CCL2, rhMCP2/CCL8, rhMCP3/CCL7, rhM-CSF, rhMIP-1α/CCL3, rhMIP-4/CCL18, rhNAP-2/CXCL7, rhNeuregulin-1α, rhNeuregulin-1β, rhOSM209aa, rhPDGF-AA, rhPDGF-BB, rhPF4V1/CXCL4, rhRANTES/CCL5, rhsCD40L/CD154, rhSCF, rhSDF-1α/CXCL12α, rhSDF-1β/CXCL 12b, rhTARC/CCL17, rhTGFβ3, rhTNF-α, rhTRAIL/CD253, rhTSLP, rhVEGF-A, rhVEGF-121. rhIL-18 from MBL (MBL International, Woburn, MA, USA), rhIL-33 from GIBCO (Invitrogen Corporation, Camarillo, CA, USA).

### 4.3. Analysis of the Sensitivity to Cleavage by hNE and hPR3

The cytokines and chemokines were dissolved in PBS or sterile water according to the recommendations of the supplier and diluted to an approximate concentration of 0.13 µg/µL. Subsequently, 9 µL (~1.2 µg) of the cytokine was mixed with 2 µL of hNE (~26 ng) or hPR3 (~340 ng) and incubated for 2.5 h at 37 °C. Two µL of sterile water was used as control. After incubation, the reactions were stopped with the addition of 3 µL of 4× sample buffer. β-Mercaptoethanol (0.5 µL) was then added to each sample followed by heating for 7 min at 85 °C. The reaction mixtures were then analyzed on 4–12% pre-cast SDS-PAGE gels (Novex, Invitrogen). To visualize the proteins, the gels were stained overnight in colloidal Coomassie staining solution (Serva Blue G) and de-stained with 25% (*v/v*) methanol in ddH_2_O for 4 h [29]. The analysis was repeated two times.

## Figures and Tables

**Figure 1 ijms-21-00651-f001:**
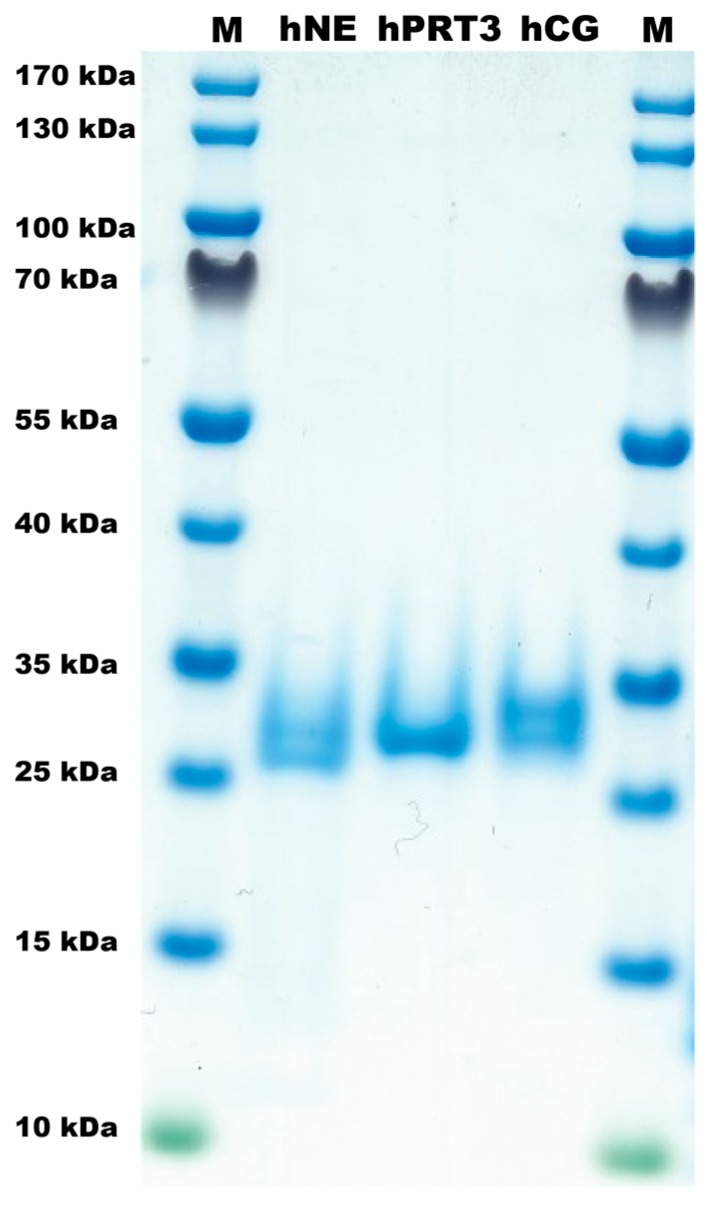
Analysis of the purified hNE, hPR-3, and hCG used in the analysis of their cleavage activity of the large panel of human cytokines and chemokines. The three human neutrophil enzymes were commercial preparations purified from peripheral blood neutrophils. The enzymes were analyzed by separation on SDS-PAGE and visualized with Coomassie Brilliant Blue staining.

**Figure 2 ijms-21-00651-f002:**
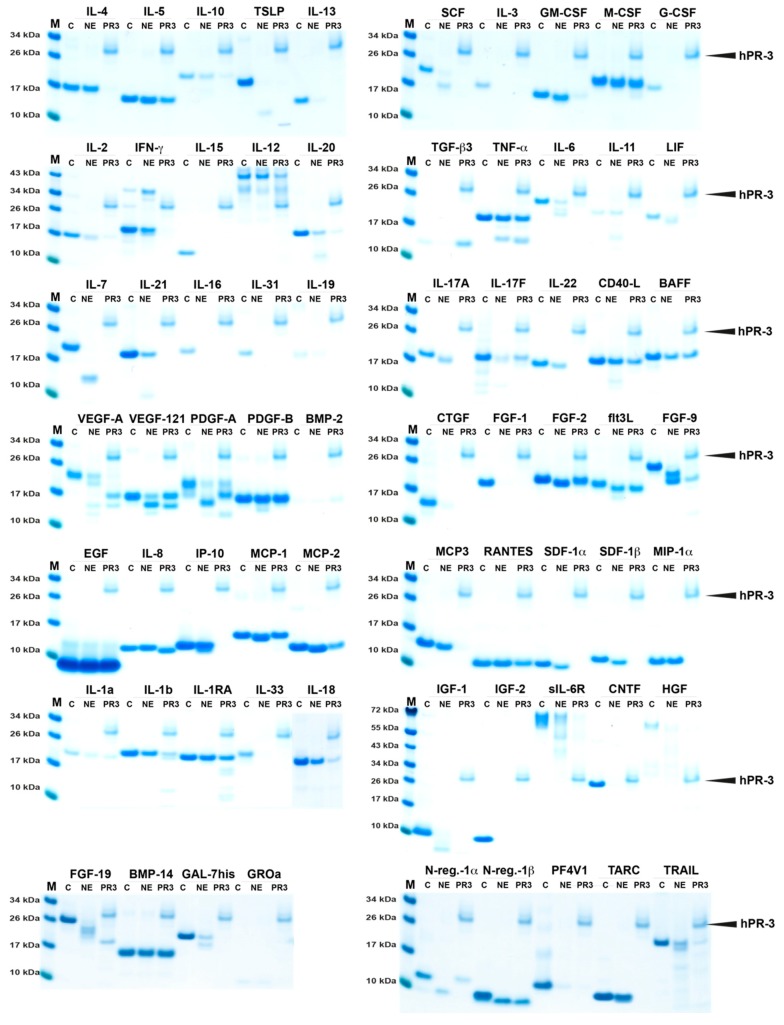
Cleavage analysis of a panel of 69 recombinant human cytokines and chemokines. Four micrograms of the various recombinant cytokines and chemokines in buffer solution were divided to cover the following experiments: one as a negative control (C) where no enzyme was added, one was cleaved with hNE and one with hPR3. The cleavage was performed at 37 °C for 2.5 h. The samples were separated on 4–12% SDS-PAGE gels under reducing conditions. Size markers are found at the left side of each gel. The gels were stained in colloidal Coomassie blue solution. Due to the relatively high concentration of hPR-3 it is visible on the gels and there marked by an arrow. The entire experiment with all the 69 cytokines and chemokines was repeated once with a new batch of cytokines and chemokines and the result was almost identical to what is shown in Figure 2.

**Figure 3 ijms-21-00651-f003:**
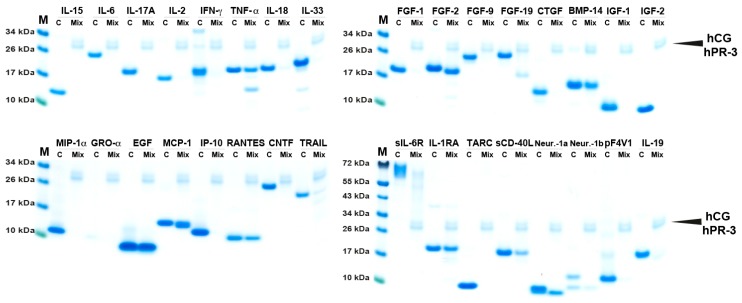
Cleavage analysis of a smaller panel of recombinant human cytokines and chemokines by a combination of three neutrophil proteases. Three micrograms of the various recombinant cytokines and chemokines in buffer solution were divided into two separate tubes. One was kept as negative control (C) where no enzyme was added, one was cleaved with a combination of three neutrophil proteases hNE, hPR3 and hCG. The cleavage was performed at 37 °C for 2.5 h. The samples were separated on 4–12% SDS-PAGE gels under reducing conditions. Size markers are found at the left side of each gel. The gels were stained in colloidal Coomassie blue solution. This analysis was performed only once.

**Figure 4 ijms-21-00651-f004:**
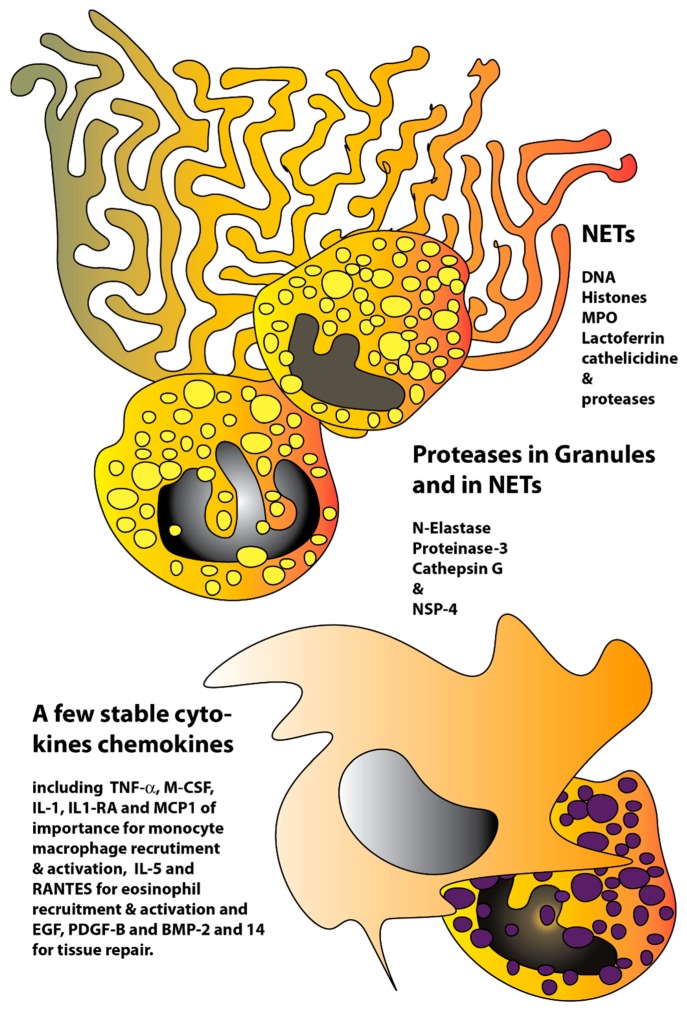
Schematic drawing of the site of infection involving neutrophils, neutrophil extracellular traps and the recruitment of monocytes/macrophages, neutrophils and possibly eosinophils. The picture shows two activated neutrophils with a neutrophil extracellular trap and an incoming monocyte-macrophage and an eosinophil. The neutrophil traps consist of neutrophil chromosomal and mitochondrial DNA to which histones and various neutrophil granule proteins are attached. In these traps, myeloperoxidase (MPO), lactoferrin, the antibacterial peptides including cathelicidine and defensins, as well as the various serine proteases have been found. The high resistance to cleavage of a few cytokines, chemokines and chemokine receptors observed (from Figure 2) indicate that these are particularly stable in order not to hamper the influx of monocytes/macrophages and possibly also eosinophils to assist the neutrophils to clear the infection.

**Table 1 ijms-21-00651-t001:** Summary of the results from the cleavage analysis of cytokines and chemokines by hNE and hPR-3 and the mix of the three neutrophil proteases hNE, hPR-3 and hCG. The analysis is based on the SDS-PAGE analysis in Figure 1 and Figure 2. (-) no cleavage activity was observed. (+) showed minor activity, (++) and (+++) partial cleavage. (++++) complete or almost complete cleaved by the enzyme. For more easy identification brackets, ¨ ¨, are inserted for the cytokines and chemokines that appear to be relatively resistant to cleavage. The results for the human mast cell chymase (HC) and human cathepsin G (hCG) are all from a previous publication [10].

Cytokine	HC	hCG	hNE	hPR3	Mix	Cytokine	hNE	hPR3	Mix
IL-4	-	-	-	++++		M-CSF	¨+¨	¨+¨	
IL-5	¨-¨	¨-¨	¨-¨	¨+¨		VEGF-121	++	+	
IL-10	-	-	+	+++		IL-16	++++	++++	
TSLP	+	-	++++	++++		FGF-9	++++	++++	++++
IL-13	+	-	+++	++++		IL-1α	¨-¨	¨-¨	
SCF	-	++++	++++	++++		IL-1RA	¨-¨	¨+¨	¨+¨
IL-3	+	++++	++++	++++		IGF-1	++++	++++	++++
GM-CSF	-	+	+	+++		IGF-2	++++	++++	++++
IL-9	-	-				sIL-6R	+	+++	+++
G-CSF	-	-	++++	++++		CNTF	++++	++++	++++
IL-2	-	+	++	+++	++++	HGF	++	+++	
IFN-γ	+	+	+	++++	++++	FGF-19	++++	++++	++++
IL-15	++++	++++	++++	++++	++++	BMP-14	¨-¨	¨-¨	¨+¨
IL-12	-	-	-	++		GAL-7H	+++	++++	
IL-20	-	-	++	+++		Neuregulin-1a	++++	+	++
IL-1β	-	-	+	++		Neuregulin-1b	++++	++++	++++
TNF-α	¨-¨	¨+¨	¨+¨	¨+¨	¨+¨	PF4V1	++++	++++	++++
IL-6	++	++++	+++	++++	++++	TARC	+	++++	++++
IL-11	+	+	+	++++		TRAIL	++	+++	+++
LIF	+	+	++	++++		BMP-2	¨-¨	¨-¨	
IL-7	-	+++	++++	++++		TGF-β3	++++	-	
IL-21	-	++	+++	++++		GROα	-	++++	++++
IL-24	-	+							
IL-31	-	++++	++++	++++					
IL-19	-	-	++	++++	++++				
IL-17A	-	++	++++	++++	++++				
IL-17F	-	-	+++	++					
IL-22	-	+	++	++++					
sCD40L	-	+	+	++	++				
BAFF	+	+	++	++					
VEGF-A	-	+++	++	++++					
BMP-7	-	+							
PDGF-A	-	-	+++	++					
PDGF-B	¨-¨	¨-¨	¨+¨	¨-¨					
IL-33	++++	++++	++++	++++	++++				
CTGF	+	+++	++++	++++	++++				
FGF-1	+	++++	++++	++++	++++				
FGF-2	+	+	++	+	++				
flt3L	++	+	++++	++++					
IL-18	+++	++++	+	+++	++++				
EGF	¨-¨	¨-¨	¨-¨	¨-¨	¨-¨				
IL-8	¨-¨	¨-¨	¨-¨	¨++¨					
IP-10	-	-	-	++++	++++				
MCP-1	¨-¨	¨-¨	+	¨-¨	¨-¨				
MCP-2	-	-	-	++					
MCP-3	-	-	+	++++					
RANTES	¨-¨	¨-¨	¨-¨	¨+¨	¨+¨				
SDF-1α	-	-	++	++++					
SDF-1β	-	+	++	++++					
MIP-1α	-	-	-	++++	++++

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
