# Peer review of "Potent and Broad but not Unselective Cleavage of Cytokines and Chemokines by Human Neutrophil Elastase and Proteinase 3"

_ijms, 2020, doi:10.3390/ijms21020651_

Round 1

Reviewer 1 Report

The manuscript is well-written and suitable for publication

Author Response

No changes suggested.

Reviewer 2 Report

The manuscript by Fu et al. entitled “Potent and broad but not unselective cleavage of cytokines and chemokines by human neutrophil elastase and proteinase 3” demonstrated broad-range protease activity of human neutrophil elastase (hNE) and proteinase 3 (hPR3) to cleave cytokines with several exceptions. First, they performed in vitro enzymatic digestion of different cytokines by hNE and hPR3 purified from human blood neutrophils and discovered that some cytokines and chemokines like M-CSF, TNF-α, MCP-1, BMP-14 etc. are resistant against digestion. Next, the authors digested cytokines or chemokines with enzyme mixture containing hNE, hPR3 and cathepsin G (hCG) and demonstrated that TFN-α, MCP-1, EGF etc. were resistant to enzymatic digestion. These results may explain how neutrophilic granules elicit inflammation through their protease activity without disturbing the activity of some cytokines or chemokines.

The figures provide detailed Coomassie blue-stained SDS-PAGE, which can clearly tell the difference between cleaved cytokines and their full-length form. Nevertheless, since the quantity of substrate (cytokines and chemokines) is fixed in each reaction. It will be better to exhibit quantitative trend of each full-length cytokine bands in the gels so that the resistance between cytokines can be compared in a more accurate way.

It is recommended to specify the number of replicates of the experiments in figure legends.

In Discussion, paragraph 2, the authors described that “The high stability of the IL-1RA could also be important for similar reasons as IL-1 is one of the key cytokines in the initiation of inflammation.” This sentence is confusing because IL-1RA is perceived as an antagonist of IL-1 receptor (PMID: 20303881). It is needed to clarify that how does the stability of anti-inflammatory ligands like IL-1RA coincide with the protease resistance of other pro-inflammatory chemokines like IL-8 etc.

In Discussion, paragraph 3, the authors described that “The inactive precursors of IL-1, IL-33 and IL-36 are also present primarily in the cytoplasm of the cells, why the cells need to be damaged in order for the neutrophil proteases to be able to cleave”. This sentence may be correct for most of the IL-1 family members like IL-1 and IL-18 but will be mistaken for IL-33. IL-33 resides in the nucleus in steady-state (PMID: 17185418) and full-length IL-33 is already in its active form (PMID: 16286016 and 29247993).

Author Response

1. Concerning the quantitative trend; 

I am sorry I do not understand the question and what we are suggested to do.

2. The number of replicates have been added to the figure legends marked in red.

3. In the discussion concerning IL-1RA we have ow corrected the statement and added a section on the potential role of the combination of IL-1 and IL-1RA. Marked in red.

In the discussion concerning IL-33. We have now modified the text and added a section on the presence, activity of the IL-33 precursor and the increase in activity upon cleavage and also a reference for this statement.